A survey on zoo mortality over a 12-year period in Italy

Scaglione Frine Eleonora frineeleonora.scaglione@unito.it 1
Biolatti Cristina 2
Pregel Paola 1
Berio Enrica 1
Cannizzo Francesca Tiziana 1
Biolatti Bartolomeo 1
Bollo Enrico 1
1 Department of Veterinary Science, University of Turin , Grugliasco , Italy
2 Istituto Zooprofilattico Sperimentale del Piemonte, Liguria e Valle d’Aosta , Turin , Italy
Roberts David
Electronic publication date: 2019 Feb 6
Publication date: 2019
Volume: 7
Electronic Location ID: e6198
Received 2017 Apr 7; Accepted 2018 Dec 2
Copyright: ©2019 Scaglione et al.
Copyright year: 2019
Copyright holder: Scaglione et al.
License: This is an open access article distributed under the terms of the Creative Commons Attribution License, which permits unrestricted use, distribution, reproduction and adaptation in any medium and for any purpose provided that it is properly attributed. For attribution, the original author(s), title, publication source (PeerJ) and either DOI or URL of the article must be cited.
License URL: https://creativecommons.org/licenses/by/4.0/

Keywords: Mammals, Mortality, Pathology, Zoo animals

Funding: The authors received no funding for this work.

==============================
Background

The zoo is a unique environment in which to study animals. Zoos have a long history of research into aspects of animal biology, even if this was not the primary purpose for which they were established. The data collected from zoo animals can have a great biological relevance and it can tell us more about what these animals are like outside the captive environment. In order to ensure the health of all captive animals, it is important to perform a post-mortem examination on all the animals that die in captivity.

Methods

The causes of mortality of two hundred and eighty two mammals which died between 2004 and 2015 in three different Italian zoos (a Biopark, a Safari Park and a private conservation center) have been investigated.

Results

Post mortem findings have been evaluated reporting the cause of death, zoo type, year and animal category. The animals frequently died from infectious diseases, in particular the causes of death in ruminants were mostly related to gastro-intestinal pathologies. pulmonary diseases were also very common in each of the zoos in the study. Moreover, death was sometimes attributable to traumas, as a result of fighting between conspecifics or during mating. Cases of genetic diseases and malformations have also been registered.

Discussion

This research was a confirmation of how conservation, histology and pathology are all connected through individual animals. These areas of expertise are extremely important to ensure the survival of rare and endangered species and to learn more about their morphological and physiological conditions. They are also useful to control pathologies, parasites and illnesses that can have a great impact on the species in captivity. Finally, this study underlines the importance of a close collaboration between veterinarians, zoo biologists and pathologists. Necropsy findings can help conservationists to determine how to support wild animal populations.

Introduction

Zoos have always been considered as establishments where wild animals are kept for exhibition (other than a circus or a pet shop) to which members of the public have access, with or without charge for admission, for a minimum period of seven calendar days per year (Hosey, Melfi & Pankhurst, 2009). Many zoos around the world keep animals confined to small spaces compared to their wide-ranging peers in the wild. Due to spatial constraints captive environments have difficulty in providing the ideal setting for natural behaviour, such as hunting, resulting in welfare issues among captive animals (Morgan & Tromborg, 2007). Sometimes, animals in captivity exhibit abnormal behaviour such stereotypies (Vaz et al., 2017) or aggressiveness (Salas et al., 2016) due to poor welfare, as behaviour is an animal’s “first line of defence” in response to environmental change, i.e., what animals do to interact with, respond to, and control their environment (Mench, 1998). Moreover, in literature, the pathologies affecting captive animals have been shown to be different from the ones affecting wild populations (Seeley et al., 2016; Strong et al., 2016).

Fortunately today, the concept of zoo has changed. Many associations cooperate together to give a new point of view about zoos. It is important to highlight that zoos are not simply cages in which animals are kept prisoner, as many people believe. They should be valued for their aims and goals. One of the key goals of many captive management programs is the eventual reintroduction of species back into the wild. Zoos exhibit species to educate the public and cultivate its appreciation of conservation or research programs. Zoos offer their visitors “edu-trainment” through shows, contact areas, and interactive exhibits. They also begin to reflect on the reason for their existence , along with issues related to animal welfare, such as behavior, exhibit design, and nutrition (Griffin, 1992).

There are many types of modern zoos: safari parks, conservation centers, landscape immersions, ecosystem exhibits, as well as bioparks and sustainable zoos. Research, education and conservation are functions which, in the last one hundred years or so, have been grafted onto the recreational rootstock of zoos (Robinson, 1989).

Keeping wild animals in captivity has advantages, first of all, for animals (conservation can be viewed as beneficial for populations of animals, if not always for individual animals kept in captivity) and for humans as well (education, conservation, recreation and scientific discovery). Wild animals in captivity may not necessarily experience negative welfare and may, in some cases, be better off than they would be in the wild (Bostock, 1993).

Conservation of endangered species is now one of the major goals of accredited zoos. The emphasis on a conservation role for zoos grew greatly in importance during the 1970s and 1980s, prompted partly by the zoos themselves and partly by external pressures, such as new international treaties and national legislation (Hosey, Melfi & Pankhurst, 2009). Another important aspect related to conservation is biodiversity.

Today, the term “conservation” and “biodiversity” are often used together, to make explicit the distinction between the conservation of living organism and non-living structures, such as buildings or books (Hosey, Melfi & Pankhurst, 2009). Another way of defining biodiversity would be as the sum total of genes, species and ecosystem in a region (WRI/IUCN/UNEP/FAO/UNESCO, 1992). The role of the zoo in the conservation of biodiversity can be defined in four general areas:

• maintenance of captive stocks of endangered species; this is the idea of zoo that can act as a kind of ‘ark’;

• support for, and practical involvement with, in situ conservation projects. Zoos could contribute to this with, amongst other things, animal planning expertise, infrastructure, and financial support;

• education and campaigning about conservation issues; this can be achieved through enclosure design, signage, keeper talks, interactive education, animal shows... Indeed, it is as important sometimes to keep species of low conservation importance in zoos as it is to keep the high-priority species, because they may be more useful in promoting the conservation message by enhancing people’s experience of animals at the zoo;

• research that benefits the science and practice of conservation; for many years, research conducted on zoo animals tended to be concerned primarily with anatomy and taxonomy, but there is a huge potential in zoo to undertake behavioral, genetic, and physiological research that contributes to the in situ and ex situ conservation of endangered species (Ryder & Feistner, 1995).

These roles and activities have been pointed out in three documents: “The World Zoo Conservation Strategy” (IUDZG/CBSG, 1993), “The World Zoo and Aquarium Conservation Strategy” (WAZA, 2005) and “Turning the Tide” (Hosey, Melfi & Pankhurst, 2009; WAZA, 2009).

The zoo is a unique environment in which to study animals. Unlike in the wild, the animals are easily accessible to the researcher, so within the framework of structured research and with the correct licenses, data from zoo animals can be collected which would otherwise be very difficult to get from their wild counterparts from a logistical point of view. Furthermore, unlike in the wild, some manipulations may be possible in the zoo to take research beyond the purely observational and into experimental approaches (Hosey, Melfi & Pankhurst, 2009), even if some data might be biased by captivity (i.e., behavior, hunting).

Zoos have a long history of research into aspects of animal biology, even if this was not the primary purpose for which they were established (Hutchins, 2001).

The data collected from zoo animals can have a greater biological relevance than data obtained from the laboratory, and it can tell us more about what these animals are like outside the captive environment (Hosey, Melfi & Pankhurst, 2009).

As a consequence, many zoos carry out their research in collaboration both with other zoos and with other bodies, such as universities and conservation agencies. Indeed, universities and zoos can complement each other, for example on topics such as the control and analysis of behavior, conservation of endangered species, the education of students and the general public (Fernandez & Timberlake, 2008). One of the greatest examples of the importance of research in zoo animals is the discovery and management of diseases.

Diseases may be ‘of concern’ to zoos either because of the direct risk of animal loss or because of the impact on the zoo of required measures in the case of an outbreak.

Each zoo will have different ‘diseases of concern’, depending on its geographical location and the types of animal in its collection, which may vary quite widely from collection to collection, and over time.

Diseases can be considered under four broad headings for all zoos:

• infectious diseases;

• degenerative diseases;

• genetic diseases;

• nutritional diseases (Hosey, Melfi & Pankhurst, 2009).

Furthermore capture, restraint, and anesthesia are also stressful procedures for animals, and particularly so for wild species. It may be better to leave an animal with a superficial injury to heal on its own without treatment if the only alternative is capture and full anesthesia. Veterinary treatment may have adverse effects on an animal’s reproductive status, or may result in aggression from conspecifics when an individual is removed for treatment and then returned into a social group. Medication that can be administered in food or drinking water may be an option when capture and injection of drug is not desirable from a welfare perspective, or when it would put veterinary staff or keepers at high risk of injury. Euthanasia is also an option (Hosey, Melfi & Pankhurst, 2009).

Preventive medicine and care play a very important role in zoos. The preventive medicine program for captive wild animals includes: stock selection, quarantine, routine health monitoring and maintenance, enclosure design, pest control, sanitation, and an employee health program. The overall goals of a preventive medicine program are to prevent disease from entering the animal collection, to ensure that the animals are properly maintained, and to avoid dissemination of diseases to other institutions, or to free-ranging populations if collection animals belong to a reintroduction program (Norton, 1993).

Preventive medicine often starts with the careful selection of new animals and a period of quarantine or isolation.

In order to protect the health of all captive animals, it is important to perform a post-mortem examination on all the animals that die in the collection and also on wild and feral animals found dead on the zoo grounds (Hosey, Melfi & Pankhurst, 2009). Many Species Survival Plans (SSPs) have extensive necropsy protocols, so the appropriate SSP Veterinary Advisor should be consulted in advance for this information (Silberman, 1988).

Proper disposal of animal carcasses is essential for both human and animal health, as well as to comply with local and federal regulations (Hinshaw, Amand & Tinkelman, 1996).

Long-term post-mortem records provide useful data on trends in health, both for individual zoos and among the wider zoo community, and this information can then help future decisions about health care in living animals.

The aim of the study was to evaluate the mortality causes, to highlight the importance of post-mortem examination and its role in preventive medicine and, secondly, to consider the importance of the veterinarian collaboration and cooperation between zoological gardens.

There are potential criticisms to this paper. Due to privacy policies, there is a lack of data regarding the animal inventory in relation to the number of necropsies. The authors are not allowed to report the data regarding the number of new animals arriving in the zoo, the number of births, the number of animals sent to other zoos, and this all influences the number of dead animals.

Materials and Methods

Sample Collection

The study on the causes of death in zoo animals was performed taking into account the years from 2004 and 2015. It was decided to focus on the Order of mammalians only, which has been divided into four categories: monogastric herbivores, ruminants, carnivores and omnivores. Two hundred and eighty two necropsies were carried out.

The animals came from three different Italian zoos (a Biopark, a Safari Park and a private conservation center) and were referred to the Department of Veterinary Science of the University of Turin (Italy).

Sample analysis

Necropsy examination was performed for each animal by two pathologists. A file was filled in with the following fields: assigned number, autopsy date, zoo of origin, species, sex, age, sampled organs.

Gross examinations were performed for each animal. Based on the macroscopic findings, the pathologists sampled organs for the histological and/or microbiological investigations.

The organs were fixed in 10% neutral buffered formalin for histological examination. The samples were paraffin-embedded and sections of 4 µm were stained with hematoxylin and eosin. Histochemical or immunohistochemical staining was performed, if necessary. All possible differential diagnoses were taken into account. Bacteriological, virological and parasitological investigations were performed, if needed.

Macroscopical and/or microscopic findings were classified according to the cause of death, including spontaneous pathology, infectious, genetic, complications (e.g., anesthesiological and surgical problems, management) and other causes (e.g., degenerative, neoplasia, nutritional and not determined diseases).

Statistical analysis

The resulting data were analyzed by GraphPad Prism (vers. 6.0; GraphPad Software, California, USA). The association between the different tested variables was assessed by χ2 Test. All results were considered statistically significant with the value p < 0.05.

Results

In Table 1 and Fig. 1, the total number of dead animals and their causes of death in the three different zoos is summarized.

Table 1 Total number of dead animals and their causes of death in the three different zoo.

Animals are classified according to their digestive system, with reference to the three zoos.

	Monogastric herbivores	Ruminants	Carnivores	Omnivores	Total	
	zoo 1	zoo 2	zoo 3	zoo 1	zoo 2	zoo 3	zoo 1	zoo 2	zoo 3	zoo 1	zoo 2	zoo 3		
Infect. diseases	19	1	11	22	75	30	1	14	2	1	1		177	
Traumas	5	3	2	6	17	6	1	4	1		1	1	47	
Complications		1		2	9	1		5			2		20	
Genetic diseases and malformations								15					15	
Other	1		2	1	4	2		10	1	1		1	23	
Tot.	25	5	15	31	105	39	2	48	4	2	4	2	282	

Figure 1 Causes of death in the three different zoos.

Dead animals classified according to their digestive system and their causes of death in the three different zoos.

Animals were classified according to their digestive system, with reference to the three zoos. Out of the 282 dead animals, 45 were monogastric herbivores, 175 were ruminants, 54 carnivores, and eight of them were omnivores.

A statistically significant association (P < 0.01) between the zoo and the category of animals was detected.

Animals were analyzed separately according to the provenance from the various zoos, and they were classified on the basis of their digestive system and the cause of death. A statistically significant association has been revealed between the category of dead animals and the three zoos (p < 0.0001). Moreover, when the zoos were considered together, a statistically significant association was also revealed between the category of dead animals and the cause of death (p < 0.0001).

In Zoo 1 out of the 60 dead animals, 25 (41.7%) were monogastric herbivores and 19 (76%) of them died from infectious diseases. Out of 31 (51.7%) ruminants, 22 (71%) died from infectious diseases. In Zoo 2, out of 162 dead animals, 105 (64.8%) were ruminants, and 75 (71.4%) died from infectious diseases, as well as 14 (29.2%) of the 48 (29.6%) carnivores. Fifteen (31.2%) carnivores died from genetic diseases or malformations and 5 (10.4%) from complications. In Zoo 3, of 60 dead animals, 30 (76.9%) of the 39 (65%) ruminants and 11 (73.3%) of the 15 (25%) monogastric herbivores died from infectious diseases.

In Zoo 1, the highest level of mortality was found in 2013, when 15 animals died (25%) and of them, 12 (80%) died from infectious diseases.

In 2015, 12 deaths were registered (20%) and of these 10 (83.3%) were from infectious diseases. Out of the 15 animals which died in 2013 in Zoo 1, 7 (46.7%) were monogastric herbivores and 7 (46.7%) were ruminants (Table 2).

Table 2 Mortality in Zoo 1 from 2005 to 2015.

Animals are classified according to their digestive system, year and cause of mortality.

	infect. disease	Traumas	Complication	Genetic diseases and malformation	Other	Total	
	Monogastric herbivores	Ruminants	Carnivores	Omnivores	Total	Monogastric herbivores	Ruminants	Carnivores	Omnivores	Total	Monogastric herbivores	Ruminants	Carnivores	Omnivores	Total	Monogastric herbivores	Ruminants	Carnivores	Omnivores	Total	Monogastric herbivores	Ruminants	Carnivores	Omnivores	Total		
2005	1				1					0					0					0				1	1	2	
2006		1			1		1			1		1			1					0					0	3	
2007	1	1			2	1	1			2					0					0		1			1	5	
2008	3	2			5					0					0					0					0	5	
2009					0			1		1					0					0					0	1	
2010		2			2	1				1					0					0					0	3	
2011		2			2	1	1			2					0					0					0	4	
2012	5	2		1	8					0					0					0					0	8	
2013	5	6	1		12	2				2		1			1					0					0	15	
2014					0		2			2					0					0					0	2	
2015	4	6			10		1			1					0					0	1				1	12	
Totale	19	22	1	1	43	5	6	1	0	12	0	2	0	0	2	0	0	0	0	0	1	1	0	1	3	60	

In 2015, out of the 12 deaths registered, 5 (41.7%) were represented by monogastric herbivores and 7 (58.3%) by ruminants. In Zoo 2 mortality was particularly high in 2009, with 32 (19.7%) deaths, 25 of which (78.1%) from infectious disease.

The most significant years for mortality in Zoo 2 were from 2006 to 2010, and involved mostly carnivores and ruminants (Table 3).

Table 3 Mortality in Zoo 2 from 2004 to 2014.

Animals are classified according to their digestive system, year and cause of mortality.

	infect. disease	Traumas	Complication	Genetic diseases and malformation	Other	Total	
	Monogastric herbivores	Ruminants	Carnivores	Omnivores	Total	Monogastric herbivores	Ruminants	Carnivores	Omnivores	Total	Monogastric herbivores	Ruminants	Carnivores	Omnivores	Total	Monogastric herbivores	Ruminants	Carnivores	Omnivores	Total	Monogastric herbivores	Ruminants	Carnivores	Omnivores	Total		
2004		12		1	13		1			1		2	1		3					0			2		2	19	
2005		1	2		3		1			1			2		2					0		3	1		4	10	
2006		4	3		7		3		1	4		1			1			7		7			1		1	20	
2007		2	4		6	2	2	1		5		2		1	3					0			1		1	15	
2008	1	5	1		7		2	1		3		2	2	1	5			6		6			1		1	22	
2009		23	2		25	1	3			4	1				1			1		1			1		1	32	
2010		13			13		2	1		3		1			1			1		1					0	18	
2011		7	1		8		1	1		2		1			1					0		1	2		3	14	
2012		7			7		2			2					0					0			1		1	10	
2013			1		1					0					0					0					0	1	
2014		1			1					0					0					0					0	1	
Totale	1	75	14	1	91	3	17	4	1	25	1	9	5	2	17	0	0	15	0	15	0	4	10	0	14	162	

The highest mortality in Zoo 3 was in 2004, with 39 (65%) deaths.

Among them, 29 (74.3%) died from infectious disease. In 2005 19 (31.7%) deaths were registered and 12 (63.1%) of them were attributable to infectious diseases.

In Zoo 3 in 2004, out of the 39 (65%) dead animals, 29 (74.3%) were ruminants and 7 (17.9%) were monogastric herbivores. In 2005, of 19 (31.7%) dead animals 10 (52.6%) were ruminants, 7 (36.8%) were monogastric herbivores, and 2 (10.5%) carnivores (Table 4).

Neoplasia, degenerative, nutritional and not determined diseases were classified as “other” in all the zoos, since some pathologies were not clearly ascribable to a specific cause (e.g., when hepatic failure occurred as a result of steatosis the primary cause of this disease could be attributable both to degenerative or a nutritional factor).

Post-mortem findings in zoos

The results obtained from laboratory investigations performed on animal death in the three zoos are reported in Tables 5–7.

Discussion

After the death of an animal, zoos are always advised to perform post-mortem examinations. The responsibility for this decision normally lies with the zoo veterinarian. Fast retrieval, storage and disposal of the carcass, contact with a specialized pathologist and record keeping are good practices to facilitate the high quality of post-mortem examinations. The safety of the staff in contact with dead animals is also relevant for inclusion in the protocol for post-mortem procedures (EU Zoo Directive, 2015).

The cause of death for each animal dying in the collection needs to be established where reasonable and practicable to do so, including, in the majority of cases, the examination of the specimen by a veterinary surgeon, pathologist or practitioner with relevant experience and training (EAZA, 2014). Often parasites, nutritional deficiencies, or dental disease, may be present in the animal collection without causing any obvious symptoms or clinical signs. Their detection at post-mortem examination frequently indicates that diagnostic tests or treatments should be performed on the remaining animals before clinical symptoms or disease transmission occur (Defra, 2012).

In this survey a general analysis has been reported, conducted by a group of veterinary pathologists, on the most common causes of death in zoo animals, over a twelve-year period. In order to provide complete and satisfactory data, 282 necropsies of zoo animals were performed.

Three different types of zoo were included in the study (a Biopark, a Safari Park and a private conservation center) as each of these zoos had a different approach to the idea of zoo animal husbandry, as described in the introduction.

Table 4 Mortality in Zoo 3 from 2004 to 2006.

Animals are classified according to their digestive system, year and cause of mortality.

	Infect. disease	Traumas	Complication	Genetic diseases and malformation	Other	Total	
	Monogastric herbivores	Ruminants	Carnivores	Omnivores	Total	Monogastric herbivores	Ruminants	Carnivores	Omnivores	Total	Monogastric herbivores	Ruminants	Carnivores	Omnivores	Total	Monogastric herbivores	Ruminants	Carnivores	Omnivores	Total	Monogastric herbivores	Ruminants	Carnivores	Omnivores	Total		
2004	6	23			29	1	3		1	5		1			1					0		2	1	1	4	39	
2005	4	7	1		12	1	3	1		5					0					0	2				2	19	
2006	1		1		2					0					0					0					0	2	
Totale	11	30	2	0	43	2	6	1	1	10	0	1	0	0	1	0	0	0	0	0	2	2	1	1	6	60	

Table 5 Results obtained from laboratory investigations performed on animal death in the zoo 1.

Register number	Year	Species	Causes of death	Lab. findings	
1A	2005	Horse	Septicemia	C.perfrigens type D	
2A	2005	Skunk	Pulmonary emphysema	–	
3A	2006	Fallow deer	Trauma	–	
4A	2006	Fallow deer	Toxemia syndrome	–	
5A	2006	Ilama	Pneumonia	–	
6A	2007	Goat	Aspiration pneumonia	–	
7A	2007	Grey squirrel	Trauma	–	
8A	2007	Deer	Trauma	–	
9A	2007	Goat	Pneumonia		
10A	2007	Patagonia hare	Septicemia	Pseudotuberculosis	
11A	2008	Ilama	Pneumonia	–	
12A	2008	Ilama	Pneumonia	–	
15 a	2008	Patagonia hare	Septicemia	–	
13A–14A	2008	Domestic rabbits	Pneumonia	–	
16A	2009	Siberian tiger	Internal hemorrhage	–	
17A	2010	Tibetan goat	Clostridial enterocolitis	Clostridiosis	
18A	2010	Hare	Trauma		
19A	2010	Tibetan goat	Septicemia	E.coli	
20A	2011	Ilama	Septicemia	Salmonellosis	
21A	2011	Antelope	Pleuritis	–	
22A	2012	Antelope	Septicemia	–	
23A	2012	Deer	Cranial trauma	–	
24A	2012	Deer	Septicemia	Actinobacillosis	
25A	2012	Hare	Trauma	–	
26A	2012	Swine	Pericarditis	–	
27–31A	2012	Hares	Pneumonia	–	
32A	2013	Deer	Septicemia	Enterococcus	
33A	2013	Ilama calf	Pneumonia	–	
34–35A	2013	Eulemurs	Trauma	–	
36A	2013	Hare	Septicemia	Pasteurella multocida	
37–40A	2013	Rabbits	Pneumonia	–	
41A	2013	Siberian tiger	Pulmonary hemorrhage	–	
42–43A	2013	Mohr gazelles	Pneumonia	–	
44A	2013	Thompson gazelle	Dystocia	–	
45–46A	2013	Deer	Pneumonia	–	
47–48A	2014	Mohr gazelle	Trauma	–	
49A	2015	Horse	Liver failure	–	
50–51A	2015	Thompson gazelle	Septicemia	–	
52A	2015	Watusi	Enteritis	–	
53A	2015	Gazelle	Pneumonia	–	
54A	2015	Yak	Pneumonia	–	
55A	2015	Goat	Trauma	–	
56A	2015	Goat	Pneumonia	–	
57–60A	2015	Rabbit	Pneumonia		

Table 6 Results obtained from laboratory investigations performed on animal death in the zoo 2.

Data	Years	Species	Causes of death	Lab. findings	
1B	2004	Lion	Neoplasia	Alveolar Carcinoma	
2B	2004	Opossum	Encephalitis	–	
3B	2004	Goat	Pneumonia	–	
4B	2004	Dromedary	Enteritis	–	
5B	2004	Antelope	Blood poisoning	–	
6B	2004	Goat	Pneumonia	–	
7B	2004	Antelope	Pneumonia	–	
8B	2004	Yak	Clostridiosis	Clostridium spp. E.coli	
9B	2004	Ilama	Thoracic Trauma	–	
10B	2004	Nilgai	Clostridiosis	Clostridium perfringens	
11B	2004	Watusi	Chronic gastritis and entheritis	–	
12B	2004	Dromedary	Septic granuloma	Trichostrongylus spp. Protostrongylus spp. Nematodirus spp.	
13B	2004	Blesbuck	Pneumonia and pleuritis	Trichostrongylus spp. Protostrongylus spp. Ostertagia spp.	
14B	2004	Eland	Blood poisoning	–	
15B	2004	Eland	Pneumonia	E.coli	
16B	2004	Lion	Paraplegia (euthanasia)	–	
17B	2004	Blesbuck	Pneumonia and pleuritis	–	
18B	2004	Goat	Pneumonia		
19B	2004	Lion	Aspiration pneumonia	–	
20B	2005	Giraffe	Heart attack	–	
21B	2005	Goat	Not determined	–	
22B	2005	Goat	Not determined	–	
23B	2005	White Lion	Aspiration pneumonia	–	
24B	2005	Lion	Neonatal mortality	–	
25B	2005	Lion	Mesothelioma	–	
26B	2005	White lion	Pneumonia	–	
27B	2005	Antelope	Severe pneumonia	–	
28B	2005	Tiger	Peritonitis	–	
29B	2005	Barbary sheep	Trauma	–	
30B	2006	Tiger	Enteritis	–	
31B	2006	Racoon	Trauma (thoracic hemorrage)	–	
32B	2006	Tiger	Not determined	–	
33B	2006	White lion	Inborn malformation	–	
34B	2006	Mouflon	Trauma	–	
35B	2006	Lion	Maxillary hypoplasia	–	
36B	2006	White Lion	Neonatal mortality	–	
37B	2006	White Lion	Neonatal mortality	–	
38B	2006	White Lion	Neonatal mortality	–	
39B	2006	White Lion	Neonatal mortality	–	
40B	2006	Waterbuck	Politrauma	–	
41B	2006	Goat	Pneumonia	–	
42B	2006	Waterbuck	Foreign body (peritonitis)	–	
43B	2006	Siberian Tiger	Severe pneumonia	–	
44B	2006	Gemsbuck (Oryx)	Pneumonia	–	
45B	2006	Waterbuck	Severe pneumonia	–	
46B	2006	Eland	Trauma	–	
47B	2006	White lion	Neonatal mortality	–	
48B	2006	White lion	Severe pneumonia	–	
49B	2007	Siberian Tiger	Severe pneumonia	–	
50B	2007	Eland	Severe pneumonia	–	
51B	2007	Racoon	Poisoning	–	
52B	2007	Hippopotamus	Trauma	–	
53B	2007	Wildebeest	Trauma	–	
54B	2007	Dromedary	Abortion	E.coli	
55B	2007	Gemsbuck (Oryx)	Trauma	–	
56B	2007	Lion	Pneumonia	–	
57B	2007	Tiger	Cranial trauma	–	
58B	2007	Tiger	Suffocation	–	
59B	2007	Tiger	Severe pneumonia	–	
60B	2007	Siberian Tiger	Severe rhinitis and pneumonia	–	
61B	2007	Gemsbuck (Oryx)	Infection	Moraxella spp.	
62B	2007	Hippopotamus	Trauma	–	
63B	2007	Buffalo	Blood poisoning	–	
64B	2008	Lion	Trauma	–	
65B	2008	Deer	Trauma	–	
66B	2008	Tiger	Internal hemmorage	–	
67B	2008	Baboon hamadryad	Hypothermia	–	
68B	2008	Buffalo	Septicemia	–	
69B	2008	White lion	Pneumonia	–	
70B	2008	Waterbuck	Hypothermia	–	
71B	2008	Gemsbuck (Oryx)	Septicemia	–	
72	2008	White Lion	Neonatal mortality	–	
73B	2008	White Lion	Neonatal mortality	–	
74B	2008	White Lion	Neonatal mortality	–	
75B	2008	Eland	Pneumonia	–	
76B	2008	Barbary sheep	Trauma	–	
77B	2008	Lion	Aspiration pneumonia	–	
78B	2008	Lion	Aspiration pneumonia	–	
79B	2008	Goat	Pneumonia	–	
80B	2008	Patagonian hare	Enteritis	–	
81B	2008	Lion	Neonatal mortality	–	
82B	2008	Lion	Neonatal mortality	–	
83B	2008	Lion	Neonatal mortality	–	
84B	2008	Eland	Severe septicemia	–	
85B	2008	Gemsbuck (Oryx)	Neonatal mortality	–	
86B	2009	Eland	Abdominal trauma	–	
87B	2009	Waterbuck	Pneumonia	E.coli	
88B	2009	Waterbuck	Trauma	–	
89B	2009	Waterbuck	Enteritis	E.coli	
90B	2009	Goat	Lymphoadenitis	–	
91B	2009	Goat	Enteritis and pneumonia	Staphylococcus xylosus Streptococcus bovis E.coli C.perfringens	
92B	2009	Goat	Enteritis	–	
93B	2009	Waterbuck	Peritonitis	–	
94B	2009	Waterbuck	Trauma	–	
95B	2009	Waterbuck	Metritis	E.coli Streptococcus bovis	
96B	2009	Tiger	Pulmonary abscess	–	
97B	2009	Tiger	Chronic nephritis	–	
98B	2009	Barbary sheep	Enteritis	Salmonella venezuelana	
99B	2009	Goat	Pneumonia	–	
100B	2009	Hippopotamus	Trauma	–	
101B	2009	Barbary sheep	Septicemia	–	
102B	2009	Barbary sheep	Enteritis	–	
103B	2009	Tibetan Goat	Enteritis	–	
104B	2009	Barbary sheep	Enteritis	–	
105B	2009	Barbary sheep	Enteritis	–	
106B	2009	Ilama	Enteritis	E.coli	
107B	2009	Dromedary	Abortion	–	
108B	2009	Lion	Neonatal mortality		
109B	2009	Barbary sheep	Deterioration	–	
110B	2009	White lion	Inborn disease (macroglossia)	–	
111B	2009	Barbary sheep calf	Enteritis and pneumonia	–	
112B	2009	Barbary sheep	Pneumonia	–	
113B	2009	Barbary sheep	Enteritis	–	
114B	2009	Goat	Pneumonia	–	
115B	2009	White donkey	Colic	–	
116B	2009	Wildebeest	Hemorragic peritonitis	–	
117B	2009	Cameroon Goat	Abortion	–	
118B	2010	Watusi	Pneumonia	–	
119B	2010	Siberian tiger	Trauma	Diaphragmatic hernia	
120B	2010	Waterbuck	Pneumonia	–	
121B	2010	Goat	Pulmonary congestion	–	
122B	2010	Goat	Pulmonary congestion	–	
123B	2010	Gemsbuck (Oryx)	Anesthesia	–	
124B	2010	Sheep	Pulmonary congestion	–	
125B	2010	Goat	Pericardial effusion	–	
126B	2010	Gemsbuck (Oryx)	Parasitic hepatitis and pneumonia	–	
127B	2010	Waterbuck calf	Neonatal mortality	–	
128B	2010	Barbary sheep	Trauma	–	
129B	2010	Siberian tiger	Fallot pentalogy	–	
130B	2010	Antelope	Hepatitis	–	
131B	2010	Gemsbuck (Oryx)	Euthanasia	Septicemia	
132B	2010	Waterbuck	Trauma	–	
133B	2010	Waterbuck	Septicemia	–	
134B	2010	Waterbuck	Septicemia	–	
135B	2010	Tibetan goat	Pericardial effusion	–	
136B	2011	Siberian tiger	Euthanasia	–	
137B	2011	Wildebeest calf	Mesenteric hemorrage	–	
138B	2011	Dromedary	Neonatal mortality	–	
139B	2011	Siberian tiger	Trauma	–	
140B	2011	Eland	Septicemia	–	
141B	2011	Gesmbuck	Trauma and septicemia	–	
142B	2011	Antelope	Not determined	–	
143B	2011	Gemsbuck	Pneumonia	–	
144B	2011	Siberian tiger	Abortion and septicemia	–	
145B	2011	Dromedary	Pulmonary congestion and septicemia	–	
146B	2011	Eland	Gastritis	–	
147B	2006	Eland	Enteritis	–	
148B	2011	Goat	Pulmonary edema	–	
149B	2011	Tiger	Not determined	–	
150B	2011	Antelope	Mycosis	–	
151B	2012	Waterbuck	Septicemia	–	
152B	2012	Waterbuck	Trauma	–	
153B	2012	Giraffe	Septicemia	Achromobacter xylosoxidans Streptococcus bovis Stenotrophomonas maltophila	
154B	2012	Cow	Septicemia	–	
155B	2012	Bison	Enteritis	–	
156B	2012	Cameroon goat	Enteritis	–	
157B	2012	Goat	Trauma	–	
158B	2012	Gemsbuck	Degradation	–	
159B	2012	Goat	Pneumonia	–	
160B	2012	Cheetah	Neoplasia	Pancreatic neoplasia	
161B	2013	Cheetah	Interstitial nephritis	–	
162B	2014	Giraffe	Pericarditis	–	

Table 7 Results obtained from laboratory investigations performed on animal death in the zoo 3.

Register number	Years	Species	Causes of death	Lab. findings	
1C	2004	Barbary sheep	Pulmunary embolism	–	
2C	2004	Ferret	Cirrhosis	–	
3C	2004	Kangaroo	Pneumonia	–	
4C	2004	Tibetan goat	Pneumonia	–	
5C	2004	Cameroon sheep	Cysticercosis	Taenia saginata	
6C	2004	Tibetan goat	Pneumonia	–	
7C	2004	Barbary sheep calf	Trauma	–	
8C	2004	Ilama	Pneumonia and pericarditis	–	
9C	2004	Kangaroo	Pneumonia	–	
10C	2004	Kangaroo	Liver disease	–	
11C	2004	Kangaroo	Pneumonia	–	
12C	2004	Crab-eating macaque	Liver failure	–	
13C	2004	Fallow deer	Pneumonia	–	
14C	2004	Fallow deer	Pneumonia	–	
15C	2004	Girgentana goat	Pneumonia	–	
16C	2004	Blackbuck	Pneumonia	–	
17C	2004	Fallow deer calf	Trauma	–	
18C	2004	Raccoon	Trauma	–	
19C	2004	Barbary sheep	Pneumonia	–	
20C	2004	Blackbuck	Pneumonia	–	
21C	2004	Tibetan goat	Pneumonia	–	
22C	2004	Barbary sheep calf	Trauma	–	
23C	2004	Tibetan goat	Pulmonary edema	–	
24C	2004	Goat	Pneumonia	–	
25C	2004	Barbary sheep	Steatosis	–	
26C	2004	Chital	Pneumonia	–	
27C	2004	Barbary sheep calf	Hemorrhagic enteritis	–	
28–29C	2004	Barbary sheep	Pneumonia	–	
30–32C	2004	Kangaroo	Pulmonary edema	–	
33C	2004	Fallow deer	Predation	–	
34C	2004	Angora Goat	Septicemia	–	
35C	2004	Blackbuck	Pneumonia	–	
36C	2004	Barbary sheep calf	Pneumonia	–	
37–39C	2004	Tibetan goat	Pneumonia	–	
40C	2005	Wallaby	Pulmonary edema	–	
41C	2005	Wallaby	Septicemia	–	
42C	2005	Squirrel	Trauma	–	
43C	2005	Ferret	Trauma	–	
44C	2005	Prairie dog	Hepatic neoplasia	–	
45C	2005	Squirrel	Pneumonia	–	
46C	2005	Ferret	Hemorrhagic enteritis	–	
47C	2005	Antelope	Pneumonia	–	
48C	2005	Barbary sheep	Trauma	–	
49C	2005	Tibetan goat	Pneumonia and pleuritis	–	
50C	2005	Kangaroo	Pericardial effusion and septicemia	–	
51C	2005	Kangaroo	Steatosis	–	
52C	2005	Barbary sheep	Pneumonia	–	
53C	2005	Goat	Trauma	–	
54C	2005	Angora goat	Pericardial effusion		
55C	2005	Fallow deer	Pneumonia	–	
56C	2005	Antelope	Peritonitis	–	
57C	2005	Dwarf goat	Trauma	–	
58C	2005	Deer	Pneumonia	–	
59C	2006	Blue monkey	Pulmonary emphysema	–	
60C	2006	Fox	Pneumonia	–	

Interesting considerations can be made, on the basis of the obtained results.

Depending on the type of zoo, the category of dead animals and causes of death were represented differently, probably due to the diverse management system of enclosures used.

Trauma can occur as a result of poor enclosure design or during capture and transport. Moreover, animals may also be injured in fights with conspecifics, particularly after introduction into a new social group, or during mating. In fact forty seven animals (16.7%) of the study died from trauma due to injuries by conspecifics or capture.

Zoo animals are protected from some health risks that are normally faced by wild animals, thanks to measures such as vaccination (Fernández-Bellon et al., 2017) and the provision of an adequate diet. At the same time, contracting an illness remains an inevitable part of zoo animal life. In fact, diseases may be spread to zoo animals through contact with conspecifics, free-ranging species, pests, such as rats and mice, keepers or visitors (Schaftenaar, 2002; Zhang et al., 2017). The study highlights that the main cause of death of captive mammals, was attributed to infectious disease (177 animals, 62.8%). Similar data were reported for each of the examined zoos and 71.7% of the examined animals which died due to infective agents were ruminants.

According to scientific literature; ruminants frequently die from infectious diseases, mostly related to their intestinal flora swing.

Links between diet and gastrointestinal problems have been reported (Zenker et al., 2009; Schilcher et al., 2013; Taylor et al., 2013). Moreover, diet and lack of structured feed items can be associated with acidosis in ruminants (Gattiker et al., 2014).

Enteritis and other pathological conditions of the digestive system were not the only diseases to have been identified, pulmonary diseases were also present. In fact, in every zoo (as described in Tables 5, 6 and 7), pneumonia and other pulmonary diseases were very common.

Respiratory infections are multifactorial diseases (Jubb, Kennedy & Palmer, 2015). Climate change is likely to be one of the factors which could increase the occurrence, distribution and prevalence of infectious diseases of the lung (Mirsaeidi et al., 2016). This result also coincides with literature, in particular for livestock. Different factors could affect livestock diseases when influenced by climate changes, such as the virulence of the pathogen itself, presence of vectors (if any), farming practices and land use, zoological and environmental factors and the establishment of new microenvironments and microclimates. The interaction of these factors is an important consideration in forecasting how livestock diseases may be spread (Gale et al., 2009).

In this study we also considered the mortality rate for each year. These data confirm that, even if there are no trigger factors of an uncontrollable epidemic in a territory, a different animal species in different years may be more prone to death.

Moreover, as demonstrated in this study, and also reported in a previous paper (Scaglione et al., 2010), in white lion cubs an increased risk of inbreeding and genetic abnormalities can be a peculiar element in zoos that are involved in the breeding of rare or endangered species, when genetic diversity can be low in captive populations (Hosey, Melfi & Pankhurst, 2009).

In Zoo 2, out of 48 dead carnivores, 14 (29.2%) died from infectious diseases and 15 (31.2%) died from genetic diseases or malformations. These latest findings, due to inbreeding, arose in felines, and in particular in the cubs. As described in the introduction, the use of studbooks may limit inbreeding and the consequent genetic abnormalities occurring in zoo animals (Leipold, 1980).

In literature different studies have been conducted on animal necropsies and they normally focus on a single animal species (EAZWV, 2008; Joyce-Zuniga et al., 2014).

A holistic approach was carried out in 1983, by the San Diego Zoo and the Department of Pathology of Zoo Animals, which conducted a survey on zoo animal necropsies over a fourteen-year period (Griner, 1983). Necropsies of wildlife and zoo animals were performed, taking into account all the species and all the taxa. The veterinarians highlighted the importance of necropsies and collection of data.

Conclusions

In conclusion, this research has been carried out to highlight how conservation, histology and pathology are:

• all connected through individual animals;

• extremely important to maintain populations of rare and endangered species and to learn more about their morphological and physiological conditions;

• useful to control diseases, parasites and illnesses which could have a great impact on those captive species. The necropsy room could represent an observatory on Zoo animal health. Finally, this study underlines the importance of:

• a close collaboration between veterinarians, zoo biologists and veterinary pathologists;

• necropsy findings which can help determine how to support wild animal populations.

Supplemental Information

Data S1 Raw data for Zoo 1

Click here for additional data file.

Data S2 Raw data for Zoo 2

Click here for additional data file.

Data S3 Raw data for Zoo 3

Click here for additional data file.

Additional Information and Declarations

Competing Interests

Author Contributions

Data Availability

The authors declare there are no competing interests.

Frine Eleonora Scaglione conceived and designed the experiments, performed the experiments, approved the final draft.

Christina Biolatti performed the experiments and analyzed the data.

Paola Pregel analyzed the data, contributed reagents/materials/analysis tools, approved the final draft.

Enrica Berio analyzed the data, prepared figures and/or tables, approved the final draft.

Francesca Tiziana Cannizzo analyzed the data, contributed reagents/materials/analysis tools, authored or reviewed drafts of the paper, approved the final draft.

Bartolomeo Biolatti conceived and designed the experiments, performed the experiments, contributed reagents/materials/analysis tools, authored or reviewed drafts of the paper, approved the final draft.

Enrico Bollo conceived and designed the experiments, performed the experiments, authored or reviewed drafts of the paper, approved the final draft.

The following information was supplied regarding data availability:

The raw data is included in Tables 5–7.

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
