# Peer review of "A survey on zoo mortality over a 12-year period in Italy"

_PeerJ, doi:10.7717/peerj.6198_

## Round 0.1 · original submission · Major Revisions

Thank you for your submission. As you see the reviewers have some major concerns that they and I believe can be rectified. My main concern is that it, while there is a lot of interesting data, there is a lack of detailed analysis such as an analysis of the source populations. Further it would be useful to know who did the examinations, who decided when and if to do diagnostic testing, and how and why causes of death diagnoses were reached.

Reviewer 1 ·

Basic reporting

• Although some sections of this paper are very well written, I feel that the authors would benefit from having a native English speaker re-read and edit the text. There are a number of sentences that do not flow well and this detracts from the content of the paper. In particular, a number of sentences throughout the abstract and results. Particular sentences in other sections include: lines 54, 98, 114,142 and 256.
• There does appear to be some bias in the introduction towards the positive aspects of zoos (specifically in lines 30, 41, 48 and 57), and although this is understandable, there needs to be more referencing and a more balanced view including the negative points of captivity to depict a more realistic view.

Experimental design

Although the idea behind this paper is a relatively good one, I feel that there is not really a specific question asked and answered.
• More detail is needed lines 166 and 167 as to which microscopical examinations were performed and their methodology.
• Clarification is needed of the term “complications” in line 170 and again in the discussion – line 284.

Validity of the findings

Within the results section, although there is a lot of information presented here it is not easy to determine the most relevant points. If the number of tables was reduced and collated into one table per zoo it would make it much easier to read through.
• I found the text in the results section rather confusing in some places e.g. line 183 – although significance is reported what is the importance of the significance? If not clarified here this needs to be further addressed in the discussion.
• The text in the results also describes results for different groups in different zoos (rather than the same groups in each zoo) which becomes confusing – it might be easier to just have one simple table showing all the groups and their causes of death from each zoo. Also, it might be easier to report where statistical significance was present rather than the areas where there was none as this again becomes confusing.
• It would also help to have a much greater number of references for the paragraph starting on line 256 as it contains many wide ranging disease statements and a large number of the conditions mentioned are not due solely to diet as seems to be implied in the text.
• Although it is interesting to link climate change to increase in infectious respiratory diseases in the paragraph beginning on line 267, the multifactorial nature of these conditions does not appear to be discussed which can appear somewhat misleading.

Additional comments

• The word “table” either needs to be standardized throughout the text to start with a t or T.
• Line 101 – this statement also needs to stress that this work can only be carried out with correct licensing.
• Line 105 – this needs to be tempered (as mentioned previously) by saying that not all information from captive animals can be applied to their wild counterparts.
• Tables 3, 4 and 5 seem to be the most useful of the group. Information needs to be collated into a smaller number of tables as it is currently complicated to cross reference between them. The Tables showing details of the post mortems are also very interesting as there are very large varieties of species in each group so it is interesting to know which species are actually represented.

Reviewer 2 ·

Basic reporting

1. Very many spelling and syntax mistakes. I suggest that you have a native English speaking colleague review your manuscript.

2. References: spelling mistakes and incorrect citation.
- line 78: WRI/MCN/UNEP/UNESCO should be WRI/IUCN/UNEP/FAO/UNESCO
- line 311: States's should be State's
- line 312: Practice should be Practise
- line 316: Johnstoni should be johnstoni / Report should be Reports
- line 328: Phatology should be Phatology / Fourteen Year should be Fourteen-year
- line 329: insert Wild Animal Park between Diego and Zoological
- line 341-343: this is an incorrect reference for line 290
- line 345: Smithsosian Institute should be Smithsonian Institution
- line 360: Startegy should be Strategy
- line 363: WRI/UNC/UNEP/FAQ/UNESCO should be WRI/IUCN/UNEP/FAO/UNESCO

3. Raw data and tables: there are many different numbers in the raw data and the tables.
Some examples:
- numbers of traumas in tables 6+12, 8+13, 10+14 are different
- numbers of zoo 1 and 2 are different in raw data and table 6 and table 8

4. Title is about a 12-year period, but line 240 says 11-year period.

5. Line 175: p<0.5 ??

6. Line 182: 96 should be 14

7. Line 201, 203, 218: monogastrics should be monogastric herbivores

Experimental design

I believe that it is more valid to consider the animal inventory in relation to the number of necropsies.
This inventory takes also into account the number of new animals arriving in the zoo, the number of births, the number of animals sent to other zoos, and this all influences the number of dead animals.

Validity of the findings

1. The numbers used in the results and discussion will change when the correct numbers are used in the tables.

2. Line 242-244: please add the type of zoo of the private collection (which type of the 6 described types on lines 62-63 ?).

3. Line 254-255: this is incorrect because in zoo 1 96% were monogastric herbivores and 70% were ruminants.

4. Line 256-263: this reference of Fox et al. 1923 is incorrect and outdated.

Additional comments

Dear colleagues,

You put a lot of work into this survey, which could be a worthwhile contribution to zoological medicine.
Please, consider all my suggestions carefully in rewriting your manuscript.

---

## Round 0.2 · Minor Revisions

Thank you for your efforts in trying to publish the raw data in this paper. I completely understand the requirement of anonymity. This however does limit, as discussed by one of the reviewers, what you can actually do with the data presented in terms of statistics. In addition further work on the manuscript is need to make it more concise in terms of wording.

Reviewer 1 ·

Basic reporting

• Although some sections of this paper have been very much improved, I feel that there are still a number of areas that would benefit from further review by a native English speaker. In particular, a number of sentences throughout the abstract and results would benefit from reassessment, e.g. line 21 and the phrase “where to study”, line 25 “pledge the health”, line 31 “died for” rather than died from which is used frequently throughout the paper- particularly in the results section. Also, lines 103, 104 and 157 which are somewhat confusing in their present form. Although I have mentioned specific sentences, the entire manuscript would benefit from a further full review to ensure all points can be clearly understood.
• It is a shame that no further data could be gathered for analysis from the three institutions, but I understand the reasons why this was not possible and the explanation in the paper is a useful addition.
• I would remove the “…” from lines 108 and 185 as these lists only require a few examples and should not be left open ended in this way.
• In a number of places there is a lack of spaces between words – this is likely just due to formatting but should be checked fully.

Experimental design

No comment

Validity of the findings

The data presented in Table 1 is also repeated in Table 2 which presents the information clearly and concisely, so I would remove Table 1 completely.
It would be helpful if some of the other tables could be combined to give a broader and more comprehensive picture rather than splitting out the data as they currently do – ie combine Tables 3 and 4 so it is possible to see what the most common causes of death were in each category for Zoo 1. The same for Tables 5 and 6 for Zoo 2 and then Tables 7 and 8 for Zoo 3. Combination of Figures 1 and 2 would also show more clearly the causes of death for each type of herbivore at each zoo.
Combining the tables in this way could also help to simplify the results section and make it easier for the reader to understand and compare between the zoos.

• Although respiratory disease has been described as multifactorial, the strong bias towards climate change as a cause is still somewhat questionable as vegetation management, species choice and issues such as stress affecting the immune system and therefore increasing the chance of respiratory disease are not mentioned at all. Particularly in Line 287 where the line “Risk assessments should focus on looking for combinations of factors that may be directly influenced by climate changes, or that may be indirectly affected through changes in human activity, such as transport and movement of animals, intensity of livestock farming and habitat change” as risk assessments should cover all aspects of particular diseases rather than focus on particular ones.

Additional comments

The authors have put a lot of work into improving the content of this paper and I feel that with a few more changes, it could be a useful addition to the literature.

Reviewer 2 ·

Basic reporting

1. Still many spelling, synthax and typing errors.
E.g. : to die of/from infection and not to die for infection, areas of expertise and not application fields, abnormal behaviour such as stereotypies and not abnormal behaviour such sterotypes, how diseases may be spread and not how diseases may be diffused, C. perfringens (in italics) and not Cl. Perfringens, E. coli (in italics) and not E.Coli, Dystocia and not Distocya, Trichostrongylus (in italics) and not Thricostrongylus, aspiration pneumonia and not Ab ingestis Pneumonia, enteritis and not entheritis, hypothermia and not ipothermia, Taenia saginata (in italics) and not TaeniaSaginata, omit rows 2007-2012 in Table 8, ...

2. Numerical errors.
Incorrect numbers in Table 1 and Table 2: total must be 282 and not 279: missing in column/row "Other" of 1 case in zoo 1 and 2 cases in zoo 3.

Experimental design

I do understand that the authors are not allowed to report the data concerning the number of animals in the zoos (inventory).
I do not agree with line 158 "There are potential criticisms in this paper". I have fundamental and existing criticism. The authors must not use statistics and percentages, because it is incorrect to compare zoos, categories and years using the bare numbers of the observations (necropsies). The authors can interpret each number only for each individual zoo, category and year. Ergo, the authors can only rewrite this manuscript as an observational report without comparison between zoos, categories and years.

Validity of the findings

I do not agree with the statistics and percentages and the interpretation of the authors (see my comment under experimental design).

Additional comments

I still feel that your observations can be a worthwhile contribution to zoo veterinary medicine.
But please, carefully rewrite your manuscript.

---

## Round 0.3 · accepted · Accept

Thank you for making the corrections. There are still a numerous formatting errors particularly where new text has been added such as double spaces or spaces not included. Please careful check the proof.

#